# The relationship between home ownership and fall-related outcomes: The National Health and Aging Trends Study

Ching-Yao Tsai[1,2,3], Tao-Hsin Tung[4], Yang-Tzu Li[5], Wei-Cheng Chen [6,7] *

**1** Department of Ophthalmology, Taipei City Hospital, Taipei, Taiwan, **2** Institute of Public Health, National Yang-Ming University, Taipei, Taiwan, **3** MS Program in Transdisciplinary Long Term Care and Bachelor's Program in Business Management, Fu Jen Catholic University, New Taipei City, Taiwan, **4** Taiwan Association of Health Industry Management and Development, Taipei, Taiwan, **5** Department of long term care, National Taipei University of Nursing and Health Science, Taipei, Taiwan, **6** Taiwan Stipendiary Co., Ltd., Kaohsiung, Taiwan, **7** Institute of Health Policy and Management, National Taiwan University, Taipei City, Taiwan

\* gevin77@hotmail.com

**Data Availability Statement:** The data that support the findings of this study are openly available in the National Health and Aging Trends Study (NHATS) at https://nhats.org/researcher/data-access/public-

## Abstract

Although many studies have tried to explore the association between fall incidents and fear of falling (FOF)/worry about fall-limited activities and various risk factors, few studies have recognized the relationship between house ownership and fall-related outcomes. The aim of this study was to assess whether house ownership will affect an older adult's experience of falling or lead to fear of falling. The National Health and Aging Trends Study (NHATS) collected data that would provide an understanding of basic trends in people aged 65 years and older living in the United States of America. This study conducted round one of the NHATS and did logistic regression to examine the relationship between house ownership and fall-related outcomes among 7,090 persons aged 65 or older. Twenty five percent of the sampled population who lacked house ownership. All fall-related outcomes (fall last month, fall last year, fear of falling, and worry about fall-limited activities) were statistically significant in the bivariate analysis. Multiple logistic regression analysis showed that house ownership (OR = 0.75, 95%CI: 0.65–0.86) was significantly associated with fear of falling after adjusting for other covariates. The findings underscore the association between the lack of house ownership and fall-related outcomes.

## Introduction

Falling has become a significant issue in older adults [1], with more than one-third of older adults experiencing a fall last year [2]. Older adults who experience falling enter a vicious cycle of physical injury, increasing dependence on daily activities, and even premature death [3]. Between 2007 and 2016 in America, fall-related deaths among older adults increased by more than 30% [4]. In addition, falling entails a high medical cost and leads to a financial burden for the U.S. government. It costs the health care system 50 billion on falls among older adults

use-files, reference number Grant number NIA U01AG032947.

**Funding:** There was no additional financial support from public or private sources.

**Competing interests:** The authors have declared that no competing interests exist.

every year [5]. With considering falls among the elderly, fear of falling should not be ignored. In recent years, many studies have examined the negative impact of fear of falling among older adults. Several studies have indicated that the elderly may develop fear of falling (FOF) after accidental falls [6, 7]. However, FOF has also been found in older adults who have never experienced falling [8]. FOF may negatively impact daily activities, and Schoene et al. indicated that a high level of FOF is associated with a low quality of life [9].

The risk factors for falling can be categorized as extrinsic and intrinsic [10]. Extrinsic factors are related to the environment of the elderly, and intrinsic factors include sociodemographic characteristics (age, race, and sex), chronic diseases (hypertension and depression), and functional status (hearing impairment and balance problem) [10, 11]. The risk factors of FOF include gender, age, physical function, mental health, comorbidities, and social network [12]. These risk factors provide useful indicators for recognizing the hazards of falling and fear of falling among older adults.

Many studies have examined the relationship between health outcomes and house ownership. People who own their houses generally have better health than those who do not [13–15]. However, little attention has been paid specifically to the association between house ownership and fall-related outcomes [16]. Environmental hazards have been recognized as a risk factor for accidental falls. However, the type of housing that is more likely to lead to falls or fear of falling among older adults has yet to be elucidated [10, 11]. The number of renter householders aged 55 or over continues to increase. In other words, nearly a quarter of all renters in the U.S. are older adults [17]. Despite extensive research on risk factors associated with falling and fear of falling, the investigation of housing ownership on fall-related outcomes is limited. Therefore, the present study examined the association between housing ownership and fall risk in older adults' residences and the residents' level of fear of falling.

## Methods

### Ethics statement

All procedures were performed in accordance with the guidelines of our institutional ethics committee and adhered to the Declaration of Helsinki. Because the data source was in the public domain and anonymized, informed consent was not given.

### Study design, setting, and sample

The data for the study were obtained from the National Health and Aging Trends Study (NHATS), which is an ongoing national program that surveys older adults aged 65 or older enrolled in Medicare [18]. The NHATS is sponsored by the National Institute on Aging (Grant number NIA U01AG032947) through a cooperative agreement with the Johns Hopkins Bloomberg School of Public Health. All procedures were performed and adhered to the tenets of the Declaration of Helsinki. All participants' information were anonymous. The NHATS data were collected annually since 2011 (Round 1), and the current study was conducted using data from Round 1. A total of 8,245 participants participated in Round 1, and this study obtained 7,090 older adults who completed the house ownership investigation. Participants who refused or did not know the answer and those who did not live at home, such as those living in retirement communities or long-term living facilities, were excluded from this study.

### Measurements

**House ownership.** According to the NHATS, participants were asked the question "Do you or your spouse/partner own the home, rent it, or is there some other arrangement?' The

study categorized the answers into "house ownership" for those who own houses and "not house ownership" for those who rent or have some other arrangement.

**Fall-related outcomes.** Many kinds of fall-related questions were asked about falling, fear of falling, and participants who worried about fall-related activities. Based on the NHATS, falls are defined as falls, slips, or trips in which participants lose their balance and land on the floor/ground or at a lower level. The participant was asked two dichotomous yes or no questions, which are "Have you fall down in the last month?" and "Do you have any experience of falling in the past year?" The participant will not be asked the question, "Do you have any experience of falling in the past year?" when their answer was "yes" to the first question. Respondents who answered "yes" in the question of "Are you worried about falling in last month?" were considered to have a fear of falling. Sample person who reported that they were worried about falling were subsequently asked the additional question, "Did you refrain from any daily activities due to the fear of falling?".

**Risk factors.** All potential risk factors for fall-related outcomes were selected from the NHATS. Furthermore, this study chose the risk factors based on previous studies that were conducted using the data from the NHATS [11, 19–21]. These factors included sociodemographic, physical medical conditions, educational level, and physical functional capacity [22].

**Sociodemographic factors.** In this study, the sociodemographic variables included were age and sex. Age was categorized into 5-year groups (65–69, 70–74, 75–79, 80–84, and ≥85 years). There were two options for the survey of gender (male/female).

**Physical medical conditions.** Based on a study by the Centers for Disease Control and Prevention, heart disease, cancer, cerebrovascular diseases, and diabetes mellitus accounted for over 60% of all deaths, with nearly 1.5 million deaths in the United States in 2001 [23]. The covariate of comorbidity in this study was presented as respondents who had more than two kinds of chronic disease, heart disease, cancer, stroke, and diabetes mellitus.

**Educational level.** The educational level of the sample was categorized into three groups: less than high school graduates, high school graduates, and higher than high school graduates.

**Physical functional capacity.** Physical function was assessed by responding to one dichotomous question (yes/no) "Did you need help with eating, getting cleaned up, using the toilet, or getting dressed in past 12 months?"

## Statistical analysis

All participants and data for this study were obtained from the first round of the NHATS (2011). The data were modified for differential nonresponses and inapplicability. Statistical analysis was performed using the Statistical Package for Social Sciences version 21.0 (SPSS Inc., Chicago, IL).

First, chi-square tests were used to examine the relationship between house ownership and fall-related outcomes. Second, multiple logistic regression analysis was used to assess fall-related outcomes and fear of falling to house ownership after adjusting for confounding factors. Statistical significance was set at P < 0.05.

## Results

The descriptive data are shown in Table 1. A total of 7,090 participants were included in this study. A total of 74.2% (5,259) were homeowners. In the study, 19.1% were aged 85 years or older, and 57.1% were male. Only 4.1% of the participants had more than two chronic diseases. In the case of education, 54.9% were high school graduates or less. Moreover, only 3% of respondents considered that they needed help in daily activities.

**Table 1. Demographics characteristics of study subjects (n = 7090).**

| Variables | n | % |
|---|---|---|
| **House owner** | | |
| No | 1831 | 25.8 |
| Yes | 5259 | 74.2 |
| **Age** | | |
| 65–69 | 1377 | 19.4 |
| 70–74 | 1521 | 21.5 |
| 75–79 | 1436 | 20.3 |
| 80–84 | 1401 | 19.8 |
| 85+ | 1355 | 19.1 |
| **Gender** | | |
| male | 3010 | 57.5 |
| female | 4080 | 42.5 |
| **Comorbidity** | | |
| Less two chronic diseases | 6778 | 95.9 |
| More than two chronic diseases | 290 | 4.1 |
| **Education** | | |
| Less than high school graduate | 1938 | 27.4 |
| high school graduate | 1943 | 27.5 |
| higher than high school graduate | 3197 | 45.2 |
| **ADL help needed** | | |
| no | 5838 | 97.0 |
| yes | 178 | 3.0 |

Table 2 presents the results of the bivariate statistical analyses of fall-related outcomes and house ownership. The relationship between fall last month and FOF/worried about fall-limited activities and house ownership was statistically significant (P<0.001).

Multiple logistic regression analysis was conducted to analyze the association between fall-related outcomes and house ownership. Compared to older adults who owned their own

**Table 2. Univariate analysis on the association between fall-related outcomes and house owner (n = 7,090).**

| | | House owner | | Unadjusted OR (95%CI) | Adjusted OR (95%CI) | P-value |
|---|---|---|---|---|---|---|
| | | Yes(n = 5259) | No(n = 1831) | | | |
| | | N(%) | N(%) | | | |
| Fall last month | Yes | 521(68.8) | 236(31.2) | 1.00 | 1.00 | <0.001 |
| | No | 4738(74.8) | 1595(25.2) | 0.74 | 0.76 | |
| | | | | (0.63–0.87) | (0.65–0.90) | |
| Fall last year | Yes | 1028(71.7) | 405(28.3) | 1.00 | 1.00 | 0.001 |
| | No | 3704(75.7) | 1188(24.3) | 0.81 | 0.85 | |
| | | | | (0.71–0.92) | (0.74–0.97) | |
| Fear of falling | Yes | 1394(67.9) | 658(32.1) | 1.00 | 1.00 | <0.001 |
| | No | 3858(76.7) | 1173(23.3) | 0.64 | 0.76 | |
| | | | | (0.57–0.72) | (0.62–0.79) | |
| Worried about fall limit activities | Yes | 576(64.4) | 319(35.6) | 1.00 | 1.00 | <0.001 |
| | No | 4682(75.6) | 1512(24.4) | 0.58 | 0.63 | |
| | | | | (0.50–0.67) | (0.54–0.73) | |

Adjustment for sex and age.

houses, those who did not were more likely to self-report FOF and worried about fall-limited activities. It was not surprising that older age, female gender, participants with more than two chronic diseases, and older adults who needed help for daily activities had a higher incidence of reporting experiencing fall accidents and FOF. The home ownership variables did not show a statistically significant association with experienced falls after adjusting for other covariates.

## Discussion

Table 2 shows that older adults that were not living in their own property were more likely to report experiencing falling accidents (falling in the last month and falling in the past year), FOF, and worry about fall-limited activities. There were three reasons that could explain the association between housing type and fall-related outcomes. First, the participant was more likely to report experiencing a falling accident, FOF, and worry about fall-limited activities because they lived in a hazardous environment. It is not surprising that tenants might be less willing to modify home hazards, which are likely to increase the risk of falling. Several environmental risk factors have been identified, such as a cluttered environment, dim room, no anti-slip material in bath/shower, without grab bars on the doorway and toilets [24]. Second, people without their own houses also suggested that they may need to move frequently and live in unfamiliar environments. It should be considered that unfamiliar places may lead to older adults experiencing falls and developing FOF [16]. Third, housing is an indirect indicator of poverty; renters are a proxy for low socioeconomic status (SES) or low income, and low incomes are highly associated with low physical activity [25, 26]. A previous study indicated that renters showed increased odds of drug and healthcare nonadherence, poor self-reported health, vascular disease, and arthritis [27]. Overall, older adults with low SES and poor health status were more likely to report experiencing falling accidents, FOF, and worry about fall-limited activities.

Tables 3 and 4 show that the relationship between fall-related outcomes and fear of falling to house ownership was analyzed through logistic regression after adjusting for covariate factors. The results of this study revealed that house ownership was associated with FOF after adjusting for other risk factors. Even though the results were not consistent with previous studies [16], the housing type among older adults should be considered as a risk factor for falls and FOF.

Age is a well-known risk factor for FOF. This study showed that with increasing age, older adults were more likely to report FOF. Even though the aging process is highly individualized, people are more vulnerable due to aging [28]. It was not surprising that people developed FOF when they were more vulnerable than they used to be.

Many studies have revealed that women are more likely to report FOF than men [12, 29]. The results of this study were also consistent with previous studies. However, the reason why FOF was more common among females still needs further research.

People diagnosed with chronic diseases were more likely to fall [6]. Likewise, FOF was more common among older adults with more chronic diseases [30, 31]. The study also indicated that comorbidity was a risk factor for FOF.

A previous study indicated that the prevalence of impairment in activities of daily living (ADL) is highly associated with age. Moreover, cognitive function is a strong predictor of ADL impairment [31]. It was not surprising that limitations in ADL were associated with the FOF. Many previous studies indicated low ADL scores as a risk factor for FOF [32–34]. The current study also showed that older adults who needed help in ADL were more likely to report FOF.

Household ownership, age, gender, comorbidity, and help needed in ADL were statistically significantly related to FOF among older adults by conducting logistic regression. To the best

**Table 3. Multiple logistic regression model of independent effort of house owner to fall related outcomes.**

| | | Fall related outcomes | | | | 95% CI | |
|---|---|---|---|---|---|---|---|
| | | Beta | SE | p value | OR | lower | upper |
| Model 1 | | | | | | | |
| Fall last month | | | | | | | |
| | **House owner** (Yes VS No) | -0.05 | 0.11 | 0.64 | 0.94 | 0.76 | 1.18 |
| | **Age(years)** | | | | | | |
| | 70-74VS 65–69 | -0.09 | 0.15 | 0.52 | 0.90 | 0.67 | 1.22 |
| | 75–79 VS 65–69 | -0.07 | 0.15 | 0.65 | 0.93 | 0.68 | 1.26 |
| | 80–84 VS 65–69 | 0.26 | 0.14 | 0.07 | 1.30 | 0.97 | 1.74 |
| | 85+ VS 65–69 | 0.36 | 0.15 | 0.01 | 1.43 | 1.06 | 1.93 |
| | **Gender** (Male VS Female) | -0.04 | 0.09 | 0.63 | 0.95 | 0.78 | 1.15 |
| | **Comorbidity** (≦2 VS >2) | 0.82 | 0.20 | <0.001 | 2.27 | 1.52 | 3.40 |
| | **Education** | | | | | | |
| | high school graduate VS Less than high school graduate | -0.40 | .129 | 0.002 | 0.66 | 0.51 | 0.85 |
| | higher than high school graduate VS Less than high school graduate | -0.38 | 0.11 | 0.001 | 0.68 | 0.54 | 0.85 |
| | **ADL help needed** (Yes VS No) | 0.38 | 0.24 | 0.11 | 1.46 | 0.90 | 2.37 |
| Model 2 | | | | | | | |
| Fall last year | | | | | | | |
| | **House owner** (Yes VS No) | -0.15 | 0.08 | 0.05 | 0.85 | 0.72 | 1.00 |
| | **Age(years)** | | | | | | |
| | 70-74VS 65–69 | -0.008 | 0.10 | 0.93 | 0.99 | 0.80 | 1.21 |
| | 75–79 VS 65–69 | 0.12 | 0.10 | 0.23 | 1.13 | 0.92 | 1.39 |
| | 80–84 VS 65–69 | 0.20 | 0.10 | 0.05 | 1.22 | 0.99 | 1.51 |
| | 85+ VS 65–69 | 0.31 | 0.11 | 0.005 | 1.37 | 1.10 | 1.71 |
| | **Gender** (Male VS Female) | -0.31 | 0.07 | <0.001 | 0.73 | 0.63 | 0.84 |
| | **Comorbidity** (≦2 VS >2) | 0.59 | 0.18 | 0.001 | 1.80 | 1.26 | 2.57 |
| | **Education** | | | | | | |
| | high school graduate VS Less than high school graduate | -0.01 | 0.09 | 0.84 | 0.98 | 0.81 | 1.18 |
| | higher than high school graduate VS Less than high school graduate | 0.15 | 0.08 | 0.07 | 1.17 | 0.98 | 1.39 |
| | **ADL help needed** (Yes VS No) | 1.06 | 0.16 | <0.001 | 2.90 | 2.09 | 4.01 |

of our knowledge, few studies have investigated the relationship between house ownership and FOF. This study suggests that house ownership should be considered a risk factor for FOF.

## Limitation

There are several limitations to this study. First, the present study used secondary data from the NHATS. NHATS data are mainly self-reported, which may result in reporting errors and recall bias. Data included many kinds of information about older adults; in other words, the data were not collected for specific purposes of the relationship between house ownership and fall-related outcomes. Second, asking the question, "In the last month, did you worry about

**Table 4. Multiple logistic regression model of house owner related to fear of falling.**

| | | Fear of falling | | | | 95% CI | |
|---|---|---|---|---|---|---|---|
| | | Beta | SE | p value | OR | lower | lower |
| Model 3 | | | | | | | |
| Fear of falling | | | | | | | |
| | **House owner** (Yes VS No) | -0.28 | 0.07 | <0.001 | 0.75 | 0.65 | 0.86 |
| | **Age(years)** | | | | | | |
| | 70-74VS 65–69 | 0.19 | 0.10 | 0.05 | 1.21 | 0.99 | 1.48 |
| | 75–79 VS 65–69 | 0.34 | 0.10 | <0.001 | 1.40 | 1.15 | 1.72 |
| | 80–84 VS 65–69 | 0.65 | 0.10 | <0.001 | 1.91 | 1.57 | 2.33 |
| | 85+ VS 65–69 | 0.97 | 0.10 | <0.001 | 2.65 | 2.17 | 3.25 |
| | **Gender** (Male VS Female) | -0.64 | 0.06 | <0.001 | 0.52 | 0.46 | 0.59 |
| | **Comorbidity** (≦2 VS >2) | 0.87 | 0.15 | <0.001 | 2.38 | 1.75 | 3.26 |
| | **Education** | | | | | | |
| | high school graduate VS Less than high school graduate | 0.05 | 0.08 | 0.48 | 1.06 | 0.89 | 1.25 |
| | higher than high school graduate VS Less than high school graduate | 0.03 | 0.07 | 0.68 | 1.03 | 0.88 | 1.20 |
| | **ADL help needed** (Yes VS No) | 1.07 | 0.15 | <0.001 | 2.93 | 2.14 | 3.99 |
| Model 4 | | | | | | | |
| Worried about fall limit activities | | | | | | | |
| | **House owner** (Yes VS No) | -0.31 | 0.10 | 0.003 | 0.73 | 0.59 | 0.89 |
| | **Age(years)** | | | | | | |
| | 70-74VS 65–69 | 0.23 | 0.15 | 0.13 | 1.26 | 0.93 | 1.72 |
| | 75–79 VS 65–69 | 0.16 | 0.16 | 0.32 | 1.17 | 0.85 | 1.62 |
| | 80–84 VS 65–69 | 0.63 | 0.15 | <0.001 | 1.88 | 1.39 | 2.55 |
| | 85+ VS 65–69 | 0.86 | 0.15 | <0.001 | 2.36 | 1.74 | 3.20 |
| | **Gender** (Male VS Female) | -0.47 | 0.09 | <0.001 | 0.62 | 0.51 | 0.75 |
| | **Comorbidity** (≦2 VS >2) | 1.15 | 0.18 | <0.001 | 3.17 | 2.18 | 4.59 |
| | **Education** | | | | | | |
| | high school graduate VS Less than high school graduate | -0.18 | 0.12 | 0.14 | 0.83 | 0.65 | 1.06 |
| | higher than high school graduate VS Less than high school graduate | -0.17 | 0.11 | 0.12 | 0.83 | 0.66 | 1.05 |
| | **ADL help needed** (Yes VS No) | 1.14 | 0.19 | <0.001 | 3.14 | 2.16 | 4.57 |

falling down?" to assess the outcomes of FOF remains controversial. Morgan et al. suggested that Falls Efficacy Scale-International (FES-I) were less misunderstood than dichotomous questions [35]. Third, NHATS data include only U.S. population, therefore the finding may be different when study in other country. We strongly suggest the research in other countries of the relationship between home ownership and fall-related outcomes are needed.

## Conclusion

Many studies have revealed that housing instability is associated with unhealthy outcomes [27, 36, 37]. The present study also showed that housing type was associated with FOF. As a result,

the government should provide low-cost rental housing for older adults in order to improve their physical and psychosocial health. Further studies are needed to evaluate the reason why older adults were more likely to develop FOF without house ownership.

## Author Contributions

**Conceptualization:** Ching-Yao Tsai, Tao-Hsin Tung, Yang-Tzu Li, Wei-Cheng Chen.

**Data curation:** Ching-Yao Tsai, Tao-Hsin Tung, Wei-Cheng Chen.

**Formal analysis:** Ching-Yao Tsai, Wei-Cheng Chen.

**Methodology:** Ching-Yao Tsai, Wei-Cheng Chen.

**Project administration:** Ching-Yao Tsai, Wei-Cheng Chen.

**Resources:** Ching-Yao Tsai, Wei-Cheng Chen.

**Supervision:** Yang-Tzu Li.

**Validation:** Ching-Yao Tsai, Wei-Cheng Chen.

**Writing – original draft:** Ching-Yao Tsai, Wei-Cheng Chen.

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
