## [Decision Letter · Decision Letter 0]

2 Aug 2021

 PGPH-D-21-00155 The relationship between home ownership and fall-related outcomes: The National Health and Aging Trends Study PLOS Global Public Health

Dear Dr. Chen,

Thank you for submitting your manuscript to PLOS Global Public Health. After careful consideration, we feel that it has merit but does not fully meet PLOS Global Public Health’s publication criteria as it currently stands. Therefore, we invite you to submit a revised version of the manuscript that addresses the points raised during the review process.

We look forward to receiving your revised manuscript.

Kind regards,

Razak M Gyasi, PhD

Academic Editor

Journal Requirements:

Additional Editor Comments (if provided):

Reviewers' comments:

Reviewer's Responses to Questions

**Comments to the Author**

1. Does this manuscript meet PLOS Global Public Health’s publication criteria? Is the manuscript technically sound, and do the data support the conclusions? The manuscript must describe methodologically and ethically rigorous research with conclusions that are appropriately drawn based on the data presented.

Reviewer #1: Yes

Reviewer #2: Yes

2. Has the statistical analysis been performed appropriately and rigorously?

Reviewer #1: I don't know

Reviewer #2: I don't know

3. Have the authors made all data underlying the findings in their manuscript fully available (please refer to the Data Availability Statement at the start of the manuscript PDF file)?

Reviewer #1: Yes

Reviewer #2: Yes

4. Is the manuscript presented in an intelligible fashion and written in standard English?

Reviewer #1: Yes

Reviewer #2: Yes

5. Review Comments to the Author

Reviewer #1: Fall risk is a major and increasing burden in aging societies. Therefore further investigation about the circumstances and reasons are interesting research topics. The question whether housing conditions are related to falls and the FOF is relevant. The secondary use of national health data is appropriate and the limitations of this approach are well discussed. The manuscript is clear and the resulting message discussed under the light of all restrictions of those statistical analyses of unspecific register data. However I would recommend to discuss another influential parameter, which is the insurance status. As Medicare covers a wide, but only selected part of the U.S. population, this certainly will create a selection bias, possibly significant in relation to housing conditions. For the international readers this should be discussed in the "limitations-section".

Otherwise thank you for the efforts to clarify knowledge on a serious medical topic!

Reviewer #2: This manuscript used the National Health and Aging Trends Study (NHATS) data to analyze the relationship between home ownership and fall-related outcomes. The findings underscore the association between the lack of house ownership and fall-related outcomes.

Questions：

1. Line 167-169, page 9: “However, the home ownership variables did not show a statistically significant association with experienced falls and worried about fall-limited activities after adjusting for other covariates. ”

This sentence seems to contradict the results in Table 4.

2. Figure 1：It is recommended to set the abscissa and ordinate title

3. Figure 1 data seems to contradict Table 2 data

4. English writing needs to be corrected.

EXPLANATION:

Line 18, page 1: Wei-Cheng Chen , MSc,

It appears that you have improperly spaced some punctuation. Consider removing a space.

Line 34, page 2: Twenty five percent of the sampled population who lacked house ownership.

It seems that there is a pronoun problem here.

6. PLOS authors have the option to publish the peer review history of their article (what does this mean?). If published, this will include your full peer review and any attached files.

**Do you want your identity to be public for this peer review?** For information about this choice, including consent withdrawal, please see our Privacy Policy.

Reviewer #1: No

Reviewer #2: No

---

## [Decision Letter · Decision Letter 1]

18 Nov 2021

The relationship between home ownership and fall-related outcomes: The National Health and Aging Trends Study

PGPH-D-21-00155R1

Dear Dr. Chen,

We're pleased to inform you that your manuscript has been judged scientifically suitable for publication and will be formally accepted for publication once it meets all outstanding technical requirements.

Within one week, you'll receive an e-mail detailing the required amendments. When these have been addressed, you'll receive a formal acceptance letter and your manuscript will be scheduled for publication.

An invoice for payment will follow shortly after the formal acceptance. To ensure an efficient process, please log into Editorial Manager at https://www.editorialmanager.com/pgph/ click the 'Update My Information' link at the top of the page, and double check that your user information is up-to-date. If you have any billing related questions, please contact our Author Billing department directly at authorbilling@plos.org.

Kind regards,

Razak M Gyasi, PhD, PD

Academic Editor

Additional Editor Comments (optional):

Reviewers' comments:

Reviewer's Responses to Questions

**Comments to the Author**

1. If the authors have adequately addressed your comments raised in a previous round of review and you feel that this manuscript is now acceptable for publication, you may indicate that here to bypass the “Comments to the Author” section, enter your conflict of interest statement in the “Confidential to Editor” section, and submit your "Accept" recommendation.

Reviewer #1: All comments have been addressed

2. Does this manuscript meet PLOS Global Public Health’s publication criteria? Is the manuscript technically sound, and do the data support the conclusions? The manuscript must describe methodologically and ethically rigorous research with conclusions that are appropriately drawn based on the data presented.

Reviewer #1: Partly

3. Has the statistical analysis been performed appropriately and rigorously?

Reviewer #1: Yes

4. Have the authors made all data underlying the findings in their manuscript fully available (please refer to the Data Availability Statement at the start of the manuscript PDF file)?

Reviewer #1: Yes

5. Is the manuscript presented in an intelligible fashion and written in standard English?

Reviewer #1: No

6. Review Comments to the Author

Reviewer #1: Questions of the review now answered, manuscript gained more clarity.

However one added sentence (Page 2 Line 12), highlighted in yellow is incomplete:

Results. Twenty five percent of the sampled population who lacked house ownership. All

13 fall-related outcomes (fall last month, fall last year, fear of falling, and worry about

Please correct, then ready from my side

7. PLOS authors have the option to publish the peer review history of their article (what does this mean?). If published, this will include your full peer review and any attached files.

**Do you want your identity to be public for this peer review?** For information about this choice, including consent withdrawal, please see our Privacy Policy.

Reviewer #1: No
